# Human T-cell activation with *Toxoplasma gondii* antigens loaded in maltodextrin nanoparticles

Laura García-López[1,3], Alejandro Zamora-Vélez[1] , Mónica Vargas-Montes[1], Juan Camilo Sanchez-Arcila[3], François Fasquelle[2], Didier Betbeder[2], Jorge Enrique Gómez-Marín[1]

Toxoplasmosis is the most prevalent parasitic zoonosis worldwide, causing ocular and neurological diseases. No vaccine has been approved for human use. We evaluated the response of peripheral blood mononuclear cells (PBMCs) to a novel construct of *Toxoplasma gondii* total antigen in maltodextrin nanoparticles (NP/TE) in individuals with varying infectious statuses (uninfected, chronic asymptomatic, or ocular toxoplasmosis). We analyzed the concentration of IFN-γ after NP/TE ex vivo stimulation using ELISA and the immunophenotypes of CD4[+] and CD8[+] cell populations using flow cytometry. In addition, serotyping of individuals with toxoplasmosis was performed by ELISA using GRA6-derived polypeptides. Low doses of NP/TE stimulation (0.9 μg NP/0.3 μg TE) achieved IFN-γ–specific production in previously exposed human PBMCs without significant differences in the infecting serotype. Increased IFN-γ expression in CD4[+] effector memory cell subsets was found in patients with ocular toxoplasmosis with NP/TE but not with TE alone. This is the first study to show how T-cell subsets respond to ex vivo stimulation with a vaccine candidate for human toxoplasmosis, providing crucial insights for future clinical trials.

## Introduction

Toxoplasmosis is a zoonotic infectious disease caused by *Toxoplasma gondii*, an obligate intracellular parasite that infects all warm-blooded animals including humans (1). This infection is considered a public health challenge, leading to several complications in immunocompromised individuals with HIV and cancer, transplant recipients, pregnant women, and newborns (2). The absence of effective therapeutic strategies for curing or preventing toxoplasmosis in humans underscores the importance of identifying vaccine agents that induce protective cellular immune responses with long-lived IFN-γ production by CD8[+] T cells (3). Over the last 30 yr, considerable efforts have been made to identify vaccine candidates, including live inactivated or attenuated parasites,

DNA, proteins, epitopes, and vector-based vaccines (4, 5). Live attenuated tachyzoites, for instance, have shown promise in inducing strong cellular immunity in animals; however, they are only available to prevent abortion in sheep and neonatal mortality in lambs (6). Despite the important development of this type of vaccine, it is not considered safe for use in humans (7).

Vaccines based on killed organisms, total lysates, and excretion/secretion products are particularly potent, activating numerous lymphocytes and eliciting robust immune responses, specifically from cytotoxic T lymphocytes (3, 4). However, the efficacy of a vaccine also depends on the immunostimulatory properties of the adjuvants and the efficiency of the delivery system for both the antigen and the adjuvant (8). Nanoparticles represent a highly promising delivery system because of their ability to protect antigens from enzymatic degradation, prolong their systemic circulation time, increase the probability of presentation to immune cells, and act as adjuvants to activate the immune system (9, 10, 11). This dual functionality allows for a reduction in the antigen dosage in vaccine formulations, thereby reducing the risk of toxicity and side effects (11, 12).

Different types of nanoparticles have been used as efficient delivery systems and stimulators of an effective immune response against *T. gondii* infection (9, 13). A novel approach using lipid-core maltodextrin nanoparticles (NP) as carriers for *T. gondii* total extract (TE) was developed several years ago (12). These nanoparticles, when conjugated with the antigen (NP/TE), have demonstrated efficacy in vitro and in vivo in intestinal and airway epithelial cell lines without toxic effects, and enabling nasal immunizations (9, 14, 15). This NP confers TE with the ability to interact directly with mucosal cells and triggers an efficient immune response against infection (10, 16). Studies in murine models have shown that NP/TE induces a specific Th1/Th17 immune response, reduces parasite load, and increases survival rate (12, 13). Furthermore, nasal immunization with NP/TE has also been explored to prevent congenital toxoplasmosis in mice (12), revealing its safety in the offspring of vaccinated infected mothers, reduced brain cysts, and reduced intraocular inflammation and ocular toxoplasmosis (17). The protection of these mice is associated with a placental cellular Th1 response, which is negatively regulated by IL-6 (17).

[1]GEPAMOL Group, Center for Biomedical Research CIBM, Faculty of Health Sciences, University of Quindío, Armenia, Colombia   [2]Vaxinano, Loos, France   [3]Department of Molecular and Cell Biology, University of California Merced, Merced, CA, USA

Correspondence: jegomez@uniquindio.edu.co

Based on these previous findings, we evaluated in vitro IFN-γ cellular responses and T-cell subpopulations of human PBMCs induced by maltodextrin nanoparticles conjugated with the total extract of *T. gondii* (NP/TE). Information on how human PBMCs respond to in vitro stimulation with NP/TE is a fundamental step in the design of future human clinical trials (18). Studies involving naïve and immune individuals to *Toxoplasma* infection provide valuable information regarding the lymphocyte populations involved in the immune response to vaccine candidates (18). In addition, we studied the immune response in chronically infected individuals according to serotype (19).

## Results

### IFN-γ production

To understand whether the clinical status associated with *T. gondii* infection affected the in vitro production of IFN-γ (the major cytokine responsible for immune protection in toxoplasmosis), we compared the PBMC IFN-γ response with TE, and NP/TE in 21 individuals: six that were uninfected with *T. gondii*, 11 chronically infected without ocular lesions, and four chronically infected with ocular lesions (Table 1). Results after stimulation with PMA/ionomycin indicated the functionality of cultured PBMCs from this group of 21 people. We used an additional control with NP alone at 10 µg/ml as a control of toxicity for high doses of NP (maltodextrin nanoparticle), where 19% (4/21) of individuals responded with the production of IFN-γ ≥ 100 pg/ml.

As expected, we observed no significant levels of IFN-γ in individuals with no previous contact with *T. gondii* after TE alone and NP/TE stimulation, and heterogeneous production in the infected asymptomatic individuals or those with ocular lesions (Fig 1). The most pronounced IFN-γ expression was detected in PBMCs from individuals with ocular manifestations after NP/TE stimulation (Fig 1). There was no significant difference in the IFN-γ levels between NP and TE at 0.5 or 0.3 µg/ml; in consequence, subsequent analysis was done for 0.3 µg/ml NP/TE.

As the results in previously infected people indicated that some people did not respond to TE or NP/TE stimuli, we calculated the percentage of people responding with a significant production of IFN-γ (≥100 pg/ml). This can provide information regarding intragroup variability in IFN-γ production. Thus, in the infected asymptomatic group stimulated with TE 0.3 µg/ml, 5 of 11 (45.4%) had significant production, in similar proportion with the conjugate NP/TE 0.3 µg/ml where 4 of 11 (36.4%) responded with significant levels. The group with a higher proportion of people responding with significant IFN-γ was the ocular toxoplasmosis group, which showed responses in 75% (3/4) with TE or NP/TE at 0.3 µg/ml.

### Flow cytometry analysis for stimulation of T-cell subsets producing IFN-γ

We then analyzed which T-cell subsets were responsible for IFN-γ production by CD4+ and CD8+ cells after ex vivo stimulation with the novel vaccine construct, NP/TE. We used the integrated mean fluorescence intensity (iMFI), defined as the multiplication of the frequency of the cell population by the gMFI, as a criterion to evaluate the IFN-γ production between groups with different disease statuses (uninfected, asymptomatic, and ocular) to TE and NP/TE. Because IFN-γ production in supernatants was higher at 0.3 µg/ml of TE or NP/TE, subsequent comparisons were made only with this concentration. We noticed slightly significantly higher IFN-γ iMFI values in CD8+ and CD4+ lymphocytes after TE, but not with the NP/TE conjugate stimuli in the asymptomatic compared with the uninfected individuals (Fig 2).

The analysis of CD4+ and CD8+ T-cell subset populations indicated higher IFN-γ production (>2 log$_{10}$ IMFI) by the CD4+Tem subset in individuals with ocular toxoplasmosis, compared with uninfected and asymptomatic individuals, in response to either TE or NP/TE at 0.3 µg/ml stimuli. Notoriously, the NP/TE conjugate, but not TE alone, increased IFN-γ expression by CD4+ Tcm population in the group of infected people with ocular lesions (Fig 3).

The differences in IFN-γ production that we found are typical of studies of immune response in human samples, revealing heterogeneous responses linked to intrinsic host genetic differences, genetic variability of the microorganism, and time of evolution of the infection. To uncover patterns within these complex data, we clustered the mean fluorescence intensity (MFI) log$_{10}$ IFN-γ–level data into a heatmap (Fig 4). The clusters revealed two large groups of effector T-cell subsets: one led by the CD4+ Tem cell subset accompanied by CD8+ Temra and CD4+ Temra, with high MFI values (upper left square), and another led by CD8+ Tem accompanied by CD8+ Tcm and CD4+ Tcm of lower intensity in MFI (upper right square). The other group was composed of most asymptomatic people with no CD4+ Tem or Temra response but a high CD8+ Tcm subset (lower left and right squares). Importantly, one group (upper left square) contained almost all individuals with ocular toxoplasmosis characterized by high IFN-γ expression (>1 log$_{10}$) in CD4+ Tem, CD4+ Temra, and CD8+ Temra subsets. In most asymptomatic and ocular toxoplasmosis individuals, the subset response was characterized by a high MFI of CD8+IFN-y, whereas most of the uninfected patients were characterized by a lower MFI of IFN-y. It is also interesting to note that the clustering pattern of the T-cell subset was more strongly related to the disease status than to the stimuli (Fig 4).

### Serotype detection by GRA6I, GRA6II, and GRA6III recombinant peptides and effect of the serotype in induction of T-cell subpopulations

In 16 individuals with IgG anti-*T. gondii*–positive results, we identified a total of 43.8% (7/16) of individuals with an undetermined serotype, whereas the other 56.2% (9/16) had an identifiable serotype. No significant differences were observed in IFN-γ expression by CD4+ or CD8+ cell subpopulations after stimulation with TE or NP/TE, according to the serotype (Table 2).

## Discussion

The present ex vivo model can be seen as a reasonable comparative picture between primary (seronegative individuals) and secondary

**Table 1. Median values and range of IFN-γ (pg/ml) in supernatants of PBMCs from individuals with different infection statuses of toxoplasmosis (negative, chronic asymptomatic, and ocular toxoplasmosis) after ex vivo culture stimulation.**

| Stimulus/infection status | Median of IFN-γ (pg/ml) (range) | | | | | | |
|---|---|---|---|---|---|---|---|
| | PMA/ionomycin | NP 10 | TE 0.3 | TE 0.5 | NP/TE 0.3 | NP/TE 0.5 | RPMI |
| Negative (n = 6) | 465 (425–474) | 0 (0–0) | 0 (0–88) | 0 (0–62) | 0 (0–0) | 0 (0–0) | 0 (0–27) |
| Asymptomatic (n = 11) | 551 (474–654) | 0 (0–4,587) | 82.8 (0–6,202) | 544 (0–30,580) | 0 (0–25,464) | 0 (0–23,823) | 0 (0–0) |
| Ocular (n = 4) | 634 (463–702) | 0 (0–0) | 3,817 (0–34,123) | 3,303 (0–29,577) | 9,494 (0–25,464) | 5,277 (0–21,931) | 0 (0–231) |

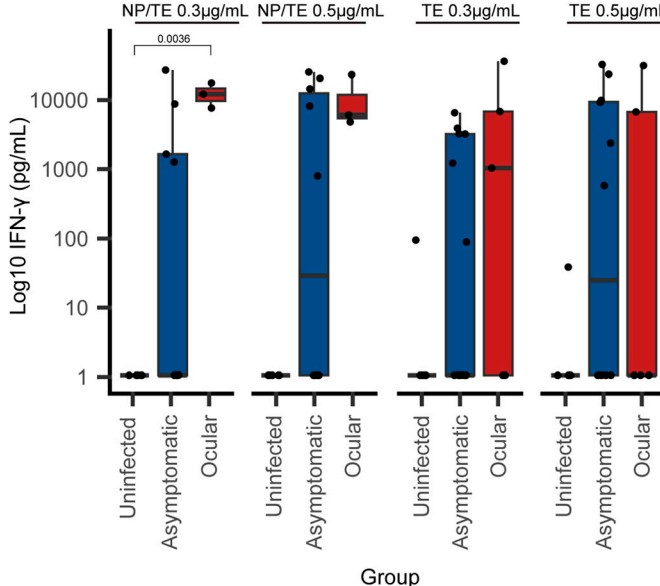

**Figure 1. Box plot of supernatant IFN-γ levels from 21 individuals with different disease statuses for toxoplasmosis (n = 4 for ocular, n = 11 for asymptomatic, and n = 6 for uninfected).**
The supernatant was collected after 72 h of ex vivo PBMC's culture stimulated with *T. gondii* total antigen at 0.3 μg/ml (TE 0.3 μg/ml) or 0.5 μg/ml (TE 0.5 μg/ml) or 0.9 μg NP conjugated to 0.3 μg/ml of *T. gondii* total antigen (NP/TE 0.3 μg/ml) or 1.5 μg NP with 0.5 μg/ml of *T. gondii* total antigen (NP/TE 0.5 μg/ml). The values plotted represent the values obtained for each stimulus minus the value obtained in the PBMCs in culture with RPMI alone from the same individual. The horizontal bar indicates the median of IFN-γ for each stimulus, and the box goes from the first quartile to the third quartile. The significance of the differences in the median across groups was calculated using the Dunn post hoc test with correction for multiple comparisons. Significant differences are marked with brackets showing *P*-values.

(asymptomatic and ocular toxoplasmosis individuals) immune responses when faced with protein extracts from a pathogen, with or without adjuvant (maltodextrin nanoparticles) in the blood (18). We found that IFN-γ was produced in response to TE or to the conjugate NP/TE in previously infected people, but not in seronegative individuals when tested in supernatants of culture. Nonsignificant differences in the IFN-γ levels were found within infected individuals (asymptomatic or with ocular lesions) in response to TE or NP/TE. However, after T-cell subpopulation analysis, we found a significant increase in CD4[+] central memory cell subsets expressing IFN-γ when nanoparticle was included (NP/TE) in the ocular toxoplasmosis group. This difference was not evident when the IMF of CD4[+] and CD8[+] T cells was analyzed probably because of the intrinsic

differences in the effector response to antigens by distinct CD8[+] and CD4[+] T effector cell types.

Our findings reveal IFN-γ production upon ex vivo NP/TE stimulation of previously exposed human PBMCs (secondary immune responses) with low doses (0.9 μg NP/0.3 μg TE), leading to a specific robust cytokine response, as demonstrated because no response in supernatants of the PBMC culture was observed from uninfected people. In contrast, when NP alone was tested at 10 μg/ml, it induced a T-cell response with high IFN-γ levels in 19% of uninfected and infected people. This can be explained by previous exposure to maltodextrin as a probiotic (20). These data are important for avoiding concentrations higher than 0.9 μg in future NP/TE clinical trials, suggesting the importance of dose optimization. This conjugate has already shown a substantial protective immune response and innocuity in animal vaccination (21), and the next steps should validate efficacy, safety, and potential protectivity in humans.

The present results showed that the NP/TE conjugate can induce an ex vivo IFN-γ response in PBMCs from infected individuals. The induction of ex vivo IFN-γ was better for the conjugate than for TE alone in seropositive individuals with ocular toxoplasmosis, suggesting that the delivery system of the maltodextrin nanoparticle can act as a scaffold for triggering specific immune response in preexposed individuals even if only PBMCs were present. Questions remain regarding how the response will occur when entry occurs by oral or nasal administration, where more efficient presenting cells reside within these tissues to process maltodextrin nanoparticles (9, 18).

We confirmed previous results showing the expression of different subsets of CD4[+] and CD8[+] cells in response to TE ex vivo stimuli in ocular and asymptomatic individuals. It is already known that genetic differences exist that can explain this difference between groups of infected people with different clinical outcomes (22). In the present study, we performed an analysis of the PBMC ex vivo response in individuals from South America, where genetic differences in the immune response were mostly linked to polymorphisms in cytokine-related genes leading to Th2-skewed response (23, 24). In the present study, individuals with ocular toxoplasmosis exhibited a distinct immune response pattern in their PBMCs compared with asymptomatic individuals. Notably, the NP/TE conjugate significantly increased the percentage of CD4[+] central memory cells in the ocular toxoplasmosis group. Previously, we demonstrated that people with ocular toxoplasmosis had a different immune response to peripheral PBMCs compared with the response of PBMCs from the asymptomatic group (23, 25). Tem CD4[+] T cells are vital for controlling *T. gondii* infection by producing cytokines, assisting in antibody production,

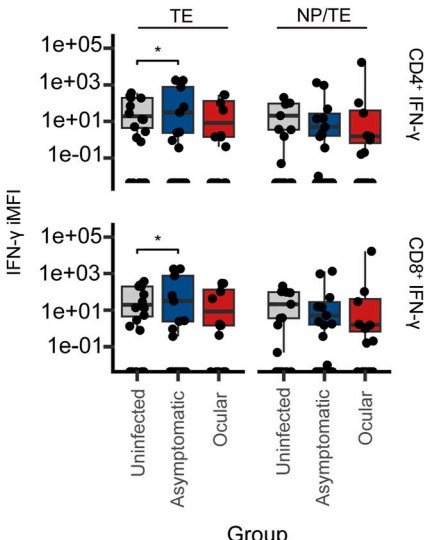

**Figure 2. Box plots of the integrated mean fluorescence intensity (iMFI) of IFN-γ in CD4⁺ and CD8⁺ lymphocytes evaluated from 23 individuals' PBMCs (n = 5 for ocular, n = 10 for asymptomatic, and n = 8 for uninfected).**
The supernatant was collected after 72 h of ex vivo culture stimulated with *T. gondii* total antigen extract 0.3 µg/ml (TE) or 0.9 µg NP conjugated to 0.3 µg/ml of *T. gondii* total antigen (NP/TE). The values in the box plots were obtained by decreasing the iMFI obtained for each stimulus minus the value obtained in the PBMCs in culture with RPMI only from the same individual. iMFI is the product value of the multiplication of the frequency of CD8⁺ or CD4⁺ T cells with the mean fluorescence intensity of IFN-γ produced by these T-cell subsets. The horizontal bar indicates the median iMFI for each stimulus, and the box goes from the first quartile to the third quartile. The significance of the differences in medians across groups was calculated using Dunn's post hoc test with correction for multiple comparisons. No significant differences were seen between the groups, except for slightly high median values after TE stimuli between uninfected and asymptomatic (*$P < 0.05$).

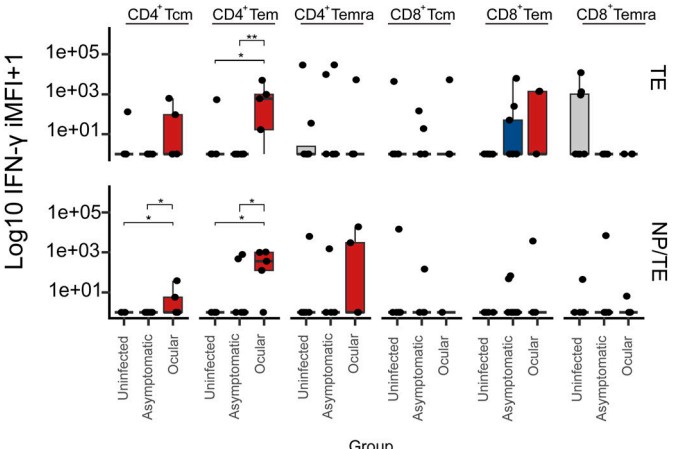

**Figure 3. Box plots of the integrated mean fluorescence intensity (iMFI) of CD4⁺ and CD8⁺ subpopulations that produce IFN-γ.**
Based on the co-expression of CD45RA and CD62L, we defined the following subpopulations for CD4⁺ and CD8⁺ lymphocytes individually: central memory subset (Tcm) CD62L⁺CD45RA⁻; effector memory subset (Tem) CD62L⁻CD45RA⁻; and effector memory cells re-expressing CD45RA subset (Temra) CD62L⁻CD45RA⁺. We evaluated PBMCs from 23 individuals with different disease statuses for toxoplasmosis as follows: ocular (n = 5), asymptomatic (n = 10), and uninfected (n = 8). All the PBMCs were stimulated for 72 h with *T. gondii* total antigen extract 0.3 µg/ml (TE) or 0.9 µg NP conjugated to 0.3 µg/ml of *T. gondii* total antigen (NP/TE). iMFI is the product of the multiplication of the frequency of CD8⁺ or CD4⁺ T cells with the MFI of IFN-γ produced by these T-cell subsets. The iMFI values represented in the box plots were obtained by decreasing each iMFI individually for each treatment from the value obtained from unstimulated cells in the RPMI medium alone. The significance of the differences in the median across groups was calculated using the Dunn post hoc test with correction for multiple comparisons. Significant differences are marked in brackets (*$P < 0.05$, **$P < 0.01$).

providing memory immune responses, and regulating the overall immune response to prevent excessive inflammation and tissue damage (26, 27). We previously demonstrated that most patients with chronic ocular toxoplasmosis have an increased membrane expression of PD1 in the central memory CD8⁺ T lymphocyte subset, leading to a total exhaustion phenotype, which can explain the defective long-term control of the parasite (28). If NP/TE can help reduce this persistent exhaustion phenomenon, further investigation into the immunomodulatory potential of NP/TE against recurrent toxoplasmosis in chronically infected individuals is required. Clinical trials including the use of NP/TE conjugates are necessary to analyze whether the effect on the induction of CD4⁺ T memory central cell subset reduces the probability of recurrence in individuals with chronic ocular toxoplasmosis by stimulating the changes in the T-cell population mediating the immune response in these individuals.

Memory T cells with different functions and phenotypes to TE and NP/TE were observed in some uninfected people (seronegative), although theoretically, this group did not have prior contact with the parasite. As they reside in endemic regions, previous contact with the parasite at low doses may explain these results. This phenomenon has also been reported in similar PBMC ex vivo experiments on *Leishmania* vaccine candidates in Brazil (29).

Significant variability in response to TE or NP/TE stimulation was observed, which was not dependent of the *Toxoplasma*-infecting serotype. We found that after NP/TE ex vivo stimuli, there was heterogeneity in IFN-γ production, which is considered the hallmark of toxoplasmosis control by the immune response (30). The variability in immune responses underscores the challenge in vaccine development, highlighting the necessity of searching dosing strategies to achieve protective immunity. Individual differences in immune responses require repeated doses in some people to obtain a protective response against many vaccines (31, 32). This is especially true for the only vaccine for malaria licensed for humans, which requires four doses to reach a poor 32% of efficacy, even though it is considered a public health utility (33). Future clinical trials in humans with NP/TE should consider this finding to evaluate the effect of the dose and repetitions necessary to reach a significant protective response.

This study substantiates the ability of the NP/TE conjugate to elicit a potent T cell–mediated immune response in human PBMCs, highlighting its potential as an effective vaccine candidate against *T. gondii*. Our results demonstrated a differential induction of T-cell subsets based on the disease status, with the NP/TE conjugate fostering a memory effector cell population in individuals with ocular toxoplasmosis. These findings merit further exploration of NP/TE as a preventive strategy for recurrent clinical toxoplasmosis.

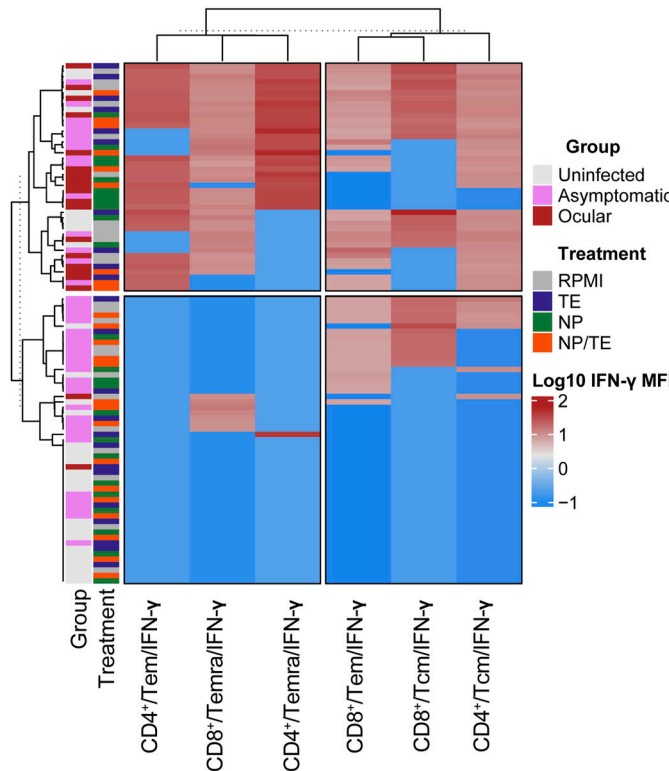

**Figure 4. Heatmap of $\log_{10}$ of IFN-γ mean fluorescence intensity values; the increase in intensity of mean fluorescence intensity is shown going from the lowest as dark blue to higher as dark brown.**

In the y-axis is shown the distribution of T-cell CD4+ and CD8+ memory subpopulations expressing IFN-γ (Tcm, Tem, and Temra). In the x-axis, the first column shows groups by Group (disease status): uninfected (n = 8); ocular (n = 5); or asymptomatic (n = 10). The second column is for the 72-h ex vivo stimulus used (Treatment): RPMI medium alone (RPMI) or after stimulation with *T. gondii* total antigen extract 0.3 µg/ml (TE) or maltodextrin nanoparticle 10 µg/ml (NP) or a conjugate of 0.9 µg NP plus 0.3 µg TE per ml (NP/TE).

## Materials and Methods

### Ethical consideration

All participants were volunteers who provided written informed consent before blood sample collection. This study was conducted in accordance with the Declaration of Helsinki. All individuals agreed to participate in the study and signed an informed consent form according to the resolution MinSalud 8430 in Colombia. The results of the clinical laboratory analysis were provided to individuals and attending physicians. This work was developed under a research contract agreement between Vaxinano and Universidad del Quindío.

### Human clinical samples

The present study was performed using serum samples and PBMCs from 23 individuals (Table 3): eight seronegative for toxoplasmosis (IgM and IgG anti-*Toxoplasma*–negative) and 16 seropositive individuals with chronic toxoplasmosis (IgM anti-*Toxoplasma*–negative and IgG anti-*Toxoplasma*–positive)

diagnosed and confirmed using the commercial kit ELFA VIDAS TOXO IgG II (Ref. 30210; bioMérieux) and TOXO IgM (Ref. 30202; bioMérieux) at the Clinical Laboratory of the Center for Biomedical Research Center (CIBM) at the University of Quindío. Among the seropositive individuals, 11 individuals were diagnosed with chronic asymptomatic toxoplasmosis, without ocular lesions, and five individuals with ocular toxoplasmosis were diagnosed as previously described (23, 28). From this group of 23 people, it is important to note that one seronegative, one asymptomatic, and one ocular were not available to measure IFN-γ levels and one asymptomatic was included to evaluate IFN-γ levels but not for phenotyping studies of T-cell populations.

### PBMC isolation and ex vivo culture

Blood samples from donors were collected in EDTA tubes (BD Vacutainer) and used to isolate PBMCs. Two PBMC isolation strategies were used to optimize sample processing, depending on the experiment. First, for IFN-γ ELISA, PBMCs were isolated by density gradient centrifugation using Histopaque-1077 (Sigma-Aldrich, Merck). First, 3 ml of Histopaque-1077 was added to 15-ml conical tubes. Next, 4 ml of blood was added slowly to avoid breaking the histopaque layer. Then, the samples were centrifuged at 450$g$ for 30 min at RT with a slow brake. The ring containing PBMCs was isolated and washed thrice with RPMI 1640 medium (Gibco, Thermo Fisher Scientific). Cells were counted using a hemocytometer and used in subsequent experiments. Second, for flow cytometry immunophenotyping, PBMCs were isolated using RBC lysis buffer (BioLegend) to ensure optimal lysis of RBCs in single-cell suspensions with minimal effects on leukocytes. We added 3 ml of blood to 30 ml of RBC 1X lysis buffer mixed separately. After incubation for 15 min at RT, the samples were centrifuged at 400$g$ for 5 min at RT, and the pellet containing the mononuclear cells was washed once with 30 ml of PBS 1X (Gibco, Thermo Fisher Scientific) and centrifuged at 400$g$ for 5 min. The remaining RBCs were lysed by adding 3 ml of ACK (ammonium–chloride–potassium; Thermo Fisher Scientific) lysis buffer, mixed, and incubated for 4 min at RT. The mononuclear cell fraction was washed with 30 ml of flow cytometry buffer (PBS, without calcium and magnesium chloride, 2% FBS, 2 mM EDTA, and 0.01% NaN$_3$) and centrifuged at 400$g$ at RT. The supernatants were discarded, and the cells were suspended in 5 ml of RPMI. All cells were counted using a hemocytometer and used in subsequent experiments.

### Stimulations for ex vivo PBMC cultures

*Toxoplasma* extract (TE) antigen was obtained after lysis of tachyzoites from a type I strain (Rh) cultured in an HFF cell line by freeze/thaw cycles, pooled, sonicated, and centrifuged as described previously (12, 16, 17, 21) at a final concentration of 0.3 or 0.5 µg/µl. Maltodextrin-based nanoparticles (NP) with an anionic phospholipid core were synthesized as previously described (12). The conjugate NP/TE was obtained by mixing 0.9 or 1.5 µg NP either with 0.3 or with 0.5 µg of TE, respectively, to reach a 3/1 weight ratio, in 24-well plates.

**Table 2. Percentage of the distribution of T-cell subsets according to the serotype (I; II*; and undetermined, UND) in the asymptomatic and ocular group of 16 individuals chronically infected with *Toxoplasma* after 72 h of ex vivo stimulation with total extract of *Toxoplasma* (TE) or conjugated TE plus maltodextrin nanoparticle (NP/TE).**

| Group | Serotype (stimulus) | CD4+% ± SD of total PBMC | CD4+% ± SD expressing IFN-γ+ | CD8+% ± SD of total PBMC | CD8% ± SD expressing IFN-γ+ |
|---|---|---|---|---|---|
| Asymptomatic (n = 11) | I (TE), n = 4 | 20.41 ± 17.06% | 41.24% ± 28.26% | 8.94% ± 6.68% | 28.47% ± 19.12% |
| | I (NP/TE), n = 4 | 20.03 ± 17.14% | 41.48% ± 27.99% | 9.02% ± 9.02% | 28.28% ± 19.39% |
| | III (TE), n = 3 | 18.44% ± 15.74% | 46.29% ± 26.79% | 8.01% ± 7.03% | 31.10% ± 18.97% |
| | III (NP/TE), n = 3 | 21.28% ± 17.1% | 41.77% ± 28% | 9.20% ± 6.84% | 29.28% ± 19.68% |
| | UND (TE), n = 4 | 22.32% ± 17.75% | 40.55% ± 28.6% | 9.14% ± 6.62% | 28.32% ± 19.63% |
| | UND (NP/TE), n = 4 | 21.28% ± 17.1% | 41.77% ± 28% | 9.20% ± 6.84% | 29.28% ± 19.68% |
| Ocular (n = 5) | I (TE), n = 1 | 60.70% | 25.90% | 14.00% | 50.30% |
| | I (NP/TE), n = 1 | 56.70% | 18.00% | 10.20%[a] | 33.60% |
| | III (TE), n = 1 | 17.40% | 23.40% | 18.70% | 43.90% |
| | III (NP/TE), n = 1 | 19.70% | 17.20% | 16.20% | 33.60% |
| | UND (TE), n = 3 | 22.37% ± 18.01% | 39.49% ± 28.98% | 9.64% ± 6.22% | 28.42% ± 19.77% |
| | UND (NP/TE), n = 3 | 23.15% ± 18.02% | 40.54% ± 29.16% | 9.93% ± 6.31% | 28.81% ± 19.83% |

[a]No serotype II was detected in this group of individuals.

**Table 3. Characteristics of individuals participating in the study.**

| Infection status | Age | Gender | *Toxoplasma* |
|---|---|---|---|
| | Median (range) | Male (n) | IgG UI/ml |
| | | Female (n) | Median (range) |
| Negative (n = 8) | 25.5 (21–34) | Male = 3 | 0 (0–0) |
| | | Female = 5 | |
| Asymptomatic (n = 11) | 40 (18–65) | Male = 6 | 119 (32–300) |
| | | Female = 5 | |
| Ocular (n = 5) | 48 (22–61) | Male = 2 | 48 (11–66) |
| | | Female = 3 | |

## Cultures for ELISA measurement of IFN-γ

To quantify IFN-γ in the culture supernatants, $1 × 10^6$ PBMCs were diluted in RPMI for stimulation. As a control for stimulation, 10 ng/ml PMA and 1 μg/ml of ionomycin calcium salt (Sigma-Aldrich) were used. After 72 h of PBMC stimulation, supernatants were collected and immediately tested. The IFN-γ concentration in the supernatants was tested in triplicate using the commercial kit Human IFN-γ Elisa Max Deluxe (Cat# 430107; BioLegend), following the manufacturer's instructions. After processing, the absorbance of the ELISA plates was read at 450 nm using an Epoch 2 spectrophotometer (BioTek Instruments). The results were processed using GraphPad Prism V. 8.0.

## Immunophenotyping

PBMCs were examined by flow cytometry to evaluate intracellular IFN-γ expression after 72 h of stimulation with the conjugate (NP/TE) and respective controls. A total of $1 × 10^6$ PBMCs per well in a 96-well plate were first incubated with 2 mmol of monensin (Sigma-Aldrich) for 2 h in the dark to trap cytokine production. Maltodextrin-based nanoparticles were tested as individual stimulus alone at 10 μg/ml, and the conjugate containing 0.9 or 1.5 μg NP either with 0.3 or with 0.5 μg of TE (NP/TE). Cells were collected and washed with flow cytometry buffer (PBS, without calcium and magnesium chloride, 2% FBS, 2 mM EDTA, and 0.01% $NaN_3$) and centrifuged at 400$g$ at RT. The cells were then incubated for 60 min on ice with the following antibodies: anti-CD3 Pe-Dazzle 594 (50 μg/ml), anti-CD4 BV510 (100 μg/ml), anti-CD8 Alexa Fluor 700 (100 μg/ml), anti-CD45RA APC/Cy7 (100 μg/ml), anti-IFN-γ Alexa Fluor 647 (25 μg/ml), and anti-CD62L BV421 (50 μg/ml). Stained PBMCs were washed twice with 100 μl of stain buffer (catalog 420201; BioLegend) and resuspended in 200 μl of the same solution. Next, permeabilization was performed using a Fix & Perm cell permeabilization kit (Cat# GAS004; Thermo Fisher Scientific) and anti-human IFN-γ-APC-Vio770 antibody was added for 30 min on ice. Cells were washed and resuspended in 300 μl FACS buffer and stored at 4°C until acquisition on an LSRFortessa (BD Biosciences). Compensation controls were

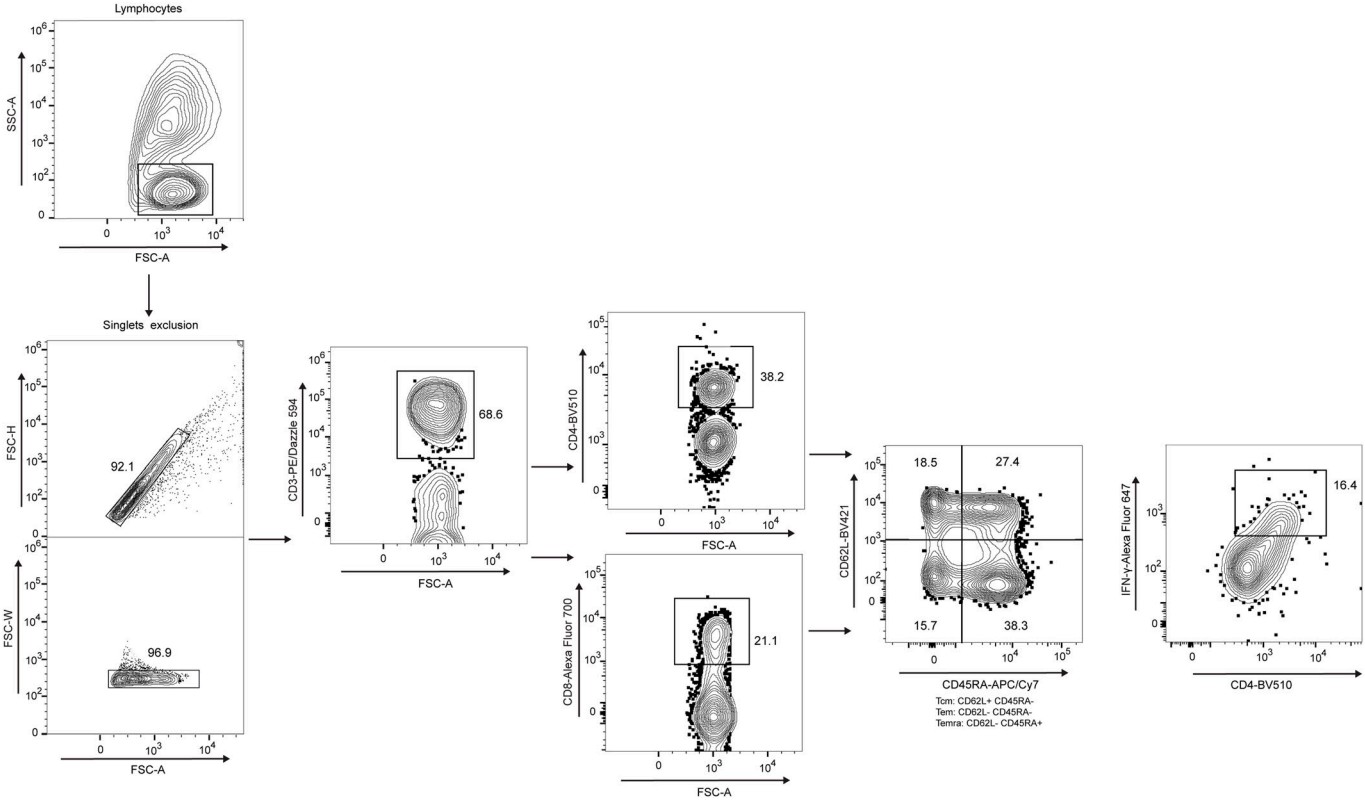

**Figure 5. Flow cytometry representative protocol.**
A gate strategy was performed as follows: to exclude cell aggregates from analyses, cells were gated on singlet region in SSC-H versus FSC-A dot plot; a second FSC-H and FSC-W versus FSC-A dot plot was created from singlet gate, and lymph region was defined; CD3⁺-positive cells were defined in CD3⁺ versus FSC-A dot plot; CD4⁺ T lymphocytes and CD8⁺ T lymphocytes were determined; CD62L versus CD45RA dot plot, gated on CD4⁺- and CD8⁺-positive T cells, respectively, was used to define Tcm, Tem, and Temra populations. From a gate on each cell population, the frequencies of IFN-γ⁺ versus CD4⁺ and CD8⁺ dot plot were determined in respective dot plots.

performed using CompBeads (BD Biosciences) that were individually stained with each antibody to calculate the compensation matrix. The gate strategy (Fig 5) included a diagram of FSC and SSC, in which the PBMC population was gated, for the subsequent selection of live PBMCs using LIVE/DEAD Fixable Aqua Dead Cell Stain Kit (Thermo Fisher Scientific). Then, CD3⁺CD8⁺ and CD3⁺CD4⁺ T cells were gated, and inside this cell population, we analyzed the subgroups of memory and activated T lymphocytes producing IFN-γ: CD4⁺ central memory subset CD3⁺CD4⁺CD45RA⁻CD62L⁺ (CD4⁺/Tcm); CD8⁺ stem cell subset CD3⁺CD8⁺CD26L⁺CD45RA⁺; CD8⁺ effector memory subset re-expressing CD45RA CD3⁺CD8⁺CD26L⁻CD45RA⁺ (CD8⁺/Temra); CD4⁺ naïve CD3⁺CD4⁺CD26L⁺CD45RA⁺; CD4⁺ effector memory subset CD4⁺CD45RA⁻ (CD4⁺/Tem); CD8⁺ effector memory subset CD8⁺CD26L⁻CD45RA⁻ (CD8⁺/Tem); and CD8⁺ central memory subset CD8⁺CD26L⁺CD45RA⁻ (CD8⁺/Tcm). The frequency of positive cells, MFI, and iMFI in arbitrary units were analyzed using FlowJo software version 10 (Tree Star). We compared the MFI or iMFI values in stimulated cells versus the RPMI-alone group without stimuli.

### Serotype detection by GRA6I, GRA6II, and GRA6III recombinant peptides by ELISA

Serotyping of volunteers with *Toxoplasma* infection was performed using an ELISA that measured specific antibodies against recombinant peptides from GRA6 donated by Dr. Corinne Mercier from Joseph Fourier University in Grenoble, France, as previously described (19). Triplicates of each serum sample were read at 450 nm using an Epoch spectrophotometer (BioTek Instruments). The positive cutoff reaction index was obtained by dividing each individual's serum absorbance by the mean absorbance plus three standard deviations (SDs) calculated from 20 serum samples from *Toxoplasma*-seronegative individuals, as described previously (19). Values ≥ 1.0 were considered as significant when positive reactivity against each peptide was detected. As a cross-reaction between type I and type III antibodies has been reported, the ratio between both types of antibodies was calculated, and the absorbance values obtained with GRA6 type I as the coating antigen were divided by the absorbance values obtained with GRA6 type III as the coating antigen, and reciprocally, as described previously (19). A positive cutoff reaction index was determined by calculating the mean average absorbance plus two SDs of serum samples from seronegative individuals. The reactivity index was calculated as the absorbance of each tested sample divided by the cutoff value. Negative reactivity thus yielded a theoretical value of ~1.0; values ≥ 1.0 were considered to indicate significant positive reactivity against the serotyping peptides, as reported previously (19). The serum groups were classified according

to the results as positive for type I, type II, or type III. If sera were negative for polypeptides, they were classified as undetermined.

## Statistical analysis

The normality of the data distribution was assessed using the Shapiro–Wilk normality test with GraphPad Prism version 5.00 for Windows (GraphPad Software, www.graphpad.com). The frequencies of cell populations, gMFI between groups, and the concentration of IFN-γ quantified by ELISA were compared using the non-parametric Dunn test with correction for multiple comparisons using the function stat_pwc from the ggpubr package (34). Descriptive statistics were applied to the serotyping and flow cytometry experiments, reporting percentages of positivity inside the individual and cell population groups and heatmaps for iMFI of the IFN-γ–producing cells.

We analyzed the grouping patterns of IFN-γ gMFI produced by a diverse set of lymphocyte populations, and if observed, grouping patterns were related to the disease status or stimulus employed in flow cytometry experiments. We built a 2-dimensional heatmap using the distance matrix built from the data ($log_{10}$-transformed) using the ComplexHeatmap package in R Statistical Language (35). Specifically, we implemented hierarchical clustering using the hclust function of the fastcluster package with ward.D2 as the clustering method and Euclidean distance (36).

# Acknowledgements

Vaxinano funded the materials and reagents for this research. M Vargas-Montes was funded with public funds through a grant of the call No 785 of 2017 from Minciencias (Colombian Ministry of Science and Technology). We thank all the individuals who donated their blood samples for this project. We also thank Dr. Corinne Mercier from the University of Grenoble for providing the antigens for the serotyping tests.

## Author Contributions

L García-López: data curation, formal analysis, investigation, visualization, methodology, and writing—original draft, review, and editing.
A Zamora-Vélez: formal analysis, investigation, visualization, and writing—original draft.
M Vargas-Montes: formal analysis and investigation.
JC Sanchez-Arcila: data curation, formal analysis, validation, visualization, and writing—original draft.
F Fasquelle: formal analysis, methodology, and writing—original draft.
D Betbeder: conceptualization, resources, funding acquisition, methodology, project administration, and writing—original draft.
JE Gómez-Marín: conceptualization, formal analysis, supervision, validation, methodology, project administration, and writing—review and editing.

## Conflict of Interest Statement

The funding was provided by Vaxinano. F Fasquelle is employed by Vaxinano. D Betbeder is CEO of Vaxinano.

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
