## [Reviewer comments · Life Science Alliance]

Life Science Alliance

Human T cell activation with *Toxoplasma gondii* antigens loaded in maltodextrin nanoparticles.

Laura Garcia-Lopez, Alejandro Zamora-Vélez, Monica Vargas-Montes, Juan Sanchez-Arcila, Francois Fasquelle, Didier Betbeder, and Jorge Gomez Marin

DOI: <https://doi.org/10.26508/lsa.202302486>

Corresponding author(s): Jorge Gomez Marin, University of Quindío

Review Timeline:

Submission Date:	2023-11-17
Editorial Decision:	2024-01-29
Revision Received:	2024-03-29
Editorial Decision:	2024-04-12
Revision Received:	2024-04-14
Accepted:	2024-04-16

Transaction Report:

January 29, 2024

Re: Life Science Alliance manuscript #LSA-2023-02486-T

Dr. Jorge Enrique Gomez Marin
University of Quindío
Centro de Investigaciones Biomedicas
Carrera 15 Calle 12 Norte
Armenia 630004

Dear Dr. Gomez Marin,

Thank you for submitting your manuscript entitled "Human T cell activation with Toxoplasma gondii antigens loaded in maltodextrin nanoparticles." to Life Science Alliance. The manuscript was assessed by expert reviewers, whose comments are appended to this letter. We invite you to submit a revised manuscript addressing the Reviewer comments.

Thank you for this interesting contribution to Life Science Alliance. We are looking forward to receiving your revised manuscript.

Sincerely,

B. MANUSCRIPT ORGANIZATION AND FORMATTING:

Reviewer #1 (Comments to the Authors (Required)):

Manuscript no: LSA-2023-02486-T

In the present manuscript Garcia-Lopez et al evaluated the efficacy of a novel nanovaccine against toxoplasmosis to induce activation and differentiation of PBMCs. The vaccine comprised of *Toxoplasma gondii* total antigen loaded in maltodextrin nanoparticle that had been previously tested in murine experimental models of toxoplasmosis. Vaccine efficacy was tested by evaluating the number of antigen-specific IFN-gamma-producing CD4 and CD8 T cells as well as the differentiation of different types of antigen-specific memory T cell, which are prerequisite for a vaccine's success. The PBMCs were isolated from different groups, i.e. uninfected, asymptomatic chronically infected and chronically infected with ocular toxoplasmosis. Authors support that the proposed vaccine raised different T cells populations in asymptomatic and people with ocular toxoplasmosis which could be characteristic of their immune background, i.e. IFN-gamma CD8 central memory T cells which are crucial in protection against parasite and CD4 central memory, respectively. thus, authors suggest that their findings provide crucial data for the design of future clinical trials, and based on this this manuscript could be considered significant and of huge interest in the era of vaccine development against toxoplasmosis.

However, authors' claims are not appropriately supported by their presented results, since the results as well as the respective figures are written in a confusing way. Moreover, some data in materials and methods as well as in results section regarding ELISA findings are missing. Thus, authors need to conduct a major revision in order to support their findings.

Specifically:

Major comments:

1. Paragraph 2.3.2: Authors should provide details for immunophenotyping which should not be limited only in staining procedure, but they should also provide details regarding culture and stimulation. E.g. Did authors used brefeldin or an analogous substance (concentration, time of incubation etc) for IFN-gamma detection? Did they use a positive control for PBMCs stimulation such as PMA-ionomycin? Were PBMCs incubated in the presence of IL-2 or anti-CD28/CD49d which helps T cells activation and differentiation? Those factors are very important for T cells activation in the absence of APCs. Also, in line 167 authors should provide information regarding the pore size of the filter used prior cells acquisition.
2. Lines 172-180: Authors should provide a representative figure of the gating strategy applied for T cells discrimination.
3. Did the authors exclude the (%), MFI and iMFI from the control T cells (namely the ones not stimulated, i.e. in medium only) in order to provide the true antigen-specific T cell populations? This should be specified in lines 180-182.
4. Paragraph 2.1 in Results section is written in a rather confused manner. The results should be presented in a more concise way, avoiding overlapping and repetitions. This could be achieved by presenting them in only one figure and not presenting five different panels which are misleading. Moreover, those panels are not described anywhere in the respective legend. I.e. the last diagram presented is the most representative and could be used for this purpose.
5. Lines 254-255: Which samples authors mean since they did not describe PMA/Ionomycin treatment in the materials and methods section nor they presented any relative result in the respective figure?
6. Lines 262-263: Provide results by giving also SD or SEM.
7. In the paragraph 2.2 of results section the figures are not the right ones for the results described. Please correct appropriately. Also, in order for authors to provide correlation among IFN-g positive T cells and disease status and explain better their findings, the most appropriate is a correlogram and not a heat-map plot. Also, this paragraph needs to be written again. Specifically, authors should present firstly (%) of CD4 and CD8 T cells to show changes among stimulations, if there are any, secondly to present the MFI and then the iMFI. This could give more information regarding the different responses among the different clinical status of the groups used.
8. Paragraph 2.3. Why authors detected T cells subpopulations in regard to different parasite serotypes? This is not clear. Also, T cells stimulation process in order to discriminate different subpopulations is not described in detail in the respective paragraph of materials and methods section. Also, have you tested IFN-gamma CD4 and CD8 T cells against serotype I, III and UND? Since this could be give more sense to the results presented in this paragraph.
9. A general comment is that T cells respond to NP10 stimulation in levels similar to antigen treatment. Have the authors exclude the spontaneous values (i.e medium only group) in order to detect the true value of response? Also, in lines 319-320, authors support that "the response observed with NP10 can be attributed to the high concentration of NP inducing cell toxicity". If so, why authors did not use lower concentrations, since the responses detected in the presence of TE also could be attributed to NP and the antigen. This should be clarified.
10. In line with the above comment, how authors explain that NP alone induce CD8 memory cells (line 336)?
11. The data show that there is not a significant difference in IFN-gamma producing T cells among naïve and infected

individuals. Moreover, there has been non-specific production of IFN-gamma when PBMCs were stimulated with NPs. How authors explain this? Is due to the antigen or the carrier dose?

12. General comment: The figures legends need to be more descriptive and the manuscript needs minor language editing.

Reviewer #2 (Comments to the Authors (Required)):

This paper shows the results of an ex vivo study on the production of IFN - gamma by peripheral blood mononuclear cells from not immune or cronically infected or with chorioretinitis due to Toxoplasma gondii individuals after stimulation with a T.gondii total antigenl loaded in maltodextrin nanoparticles NPL/TE at different concentration as a preliminary study for a vaccine . Furthermore the immunophenotyping of CD4andCD8 cells and the serotyping of infecting parasite were performed . The autografato found that the immunological response after stimulation with NPL/TE is different in different individuals and this is not a consequence of different genotype of T. Gondii infection and that problem with ocular tocoplasmosis have a different a stronger response to the antigen stimulation. They conclude that NPL/TE coniugate could be an eccellenti vaccine candidate . In my opinion the study is well conducted the results are clearly espressed but must be stressed that are really preliminar data due to the low number of test ed samples . I want also to ask to specify why the oincubatipn time eas chosen at 72 h. The results of serotyping must be described in the text also. Line 70-71 there is a tepetition

Reviewer #3 (Comments to the Authors (Required)):

In the studies described in this manuscript, the author examined IFN- production by PBMCs obtained from three groups of individuals in the conditions on infection with Toxoplasma gondii (seronegative [uninfected], chronically infected [asymptomatic], and ocular toxoplasmosis) following stimulation with medium (a control), toxoplasma extracts (TE) as a conventional antigen, nanoparticles not containing TE (NP), and nanoparticles conjugated with TE (NP/TE). However, the results look problematic. One issue is high IFN- productions following stimulation with NP. Another issue is large variations among individuals within each of the three groups of the individuals. In addition, in stimulation with NP/TE, there were no statistically significant differences in the IFN- production between the asymptomatic and ocular toxoplasmosis individuals, indicating that this assay would not be helpful for diagnosis of ocular toxoplasmosis. The authors also analyzed T cell subpopulations that produce IFN- in response to these four antigen preparations. However, there are no clear differences in the frequencies of IFN- + cells in each of 6 subpopulations of the T cells among the three groups of the individuals and among four different antigen preparations (Figure 5). Although the authors state in the Abstract that "We found the populations with higher IFN- were CD8+ central memory T cell subset in chronic asymptomatic infection and of CD4+ central memory cell and terminally differentiated effector memory subset from people with ocular toxoplasmosis", the results shown in Figure 5 do not support this statement. Overall the quality and significance of the data show in in this manuscript are not high.

Armenia, 29 March 2024

Life Science Alliance manuscript #LSA-2023-02486-T

Title: Human T cell activation with *Toxoplasma gondii* antigens loaded in maltodextrin nanoparticles.

Dr. Eric Sawey, PhD

Executive Editor

Life Science Alliance

Dear Doctor Sawey,

We have reviewed our manuscript, and our answers to specific questions and modifications are as follows.

Reviewer #1 (Comments to the Authors (required))

Major comments:

1. Paragraph 2.3.2: Authors should provide details for immunophenotyping which should not be limited only in staining procedure, but they should also provide details regarding culture and stimulation. E.g. Did authors use brefeldin or an analogous substance (concentration, time of incubation etc) for IFN-gamma detection? Did they use a positive control for PBMCs stimulation such as PMA-ionomycin? Were PBMCs incubated in the presence of IL-2 or anti-CD28/CD49d which helps T cells activation and differentiation? Those factors are very important for T cell activation in the absence of APCs. Also, in line 167 authors should provide information regarding the pore size of the filter used prior to cell acquisition.

Answer: We thank the reviewer for this remark, we inadvertently omitted this information. We used PMA stimulation as a control, which is detailed in lane 303 and

304:10 ng/mL of phorbol 12-myristate 13-acetate (PMA), and 1 µg/mL of ionomycin calcium salt (Sigma-Aldrich, France) as controls. The cells were incubated with monensin for 2 h in the dark as a secretion inhibitor to retain intracellular cytokines and stained. This was included in Method section, lane 314. Because we cultured total PBMCs without selecting specific cell types, we had only a small portion of APCs. Moreover, this strategy of using nanoparticles allows for antigen release and T cell activation. We did not have the opportunity to explore the mechanism of antigen presentation and stimulation of T cells by these nanoparticles. The pore size of the filter was 3 µm.

2. Lines 172-180: Authors should provide a representative figure of the gating strategy applied for T cells discrimination.

Answer: We have included a new figure (Figure 1) with the gating strategy.

3. Did the authors exclude the (%), MFI and iMFI from the control T cells (namely the ones not stimulated, i.e. in medium only) in order to provide the true antigen-specific T cell populations? This should be specified in lines 180-182.

Answer: We did not subtract the %, MFI, or iMFI values. This is because we consider it important to show that, even without stimuli, some patients, such as those with ocular toxoplasmosis, can have naturally increased levels of IFN-γ. Therefore, we used the RPMI group for all analyses. It is important to show that, even without stimuli, some patients, such as those with ocular toxoplasmosis, can have naturally increased levels of IFN-γ. To address this, we remade the plots by decreasing the RPMI values for %, MFI, and iMFI (explained lanes 344 and 345).

4. Paragraph 2.1 in the Results section is written in a rather confused manner. The results should be presented in a more concise way, avoiding overlapping and repetitions. This could be achieved by presenting them in only one figure and not presenting five different panels which are misleading. Moreover, those panels are not described anywhere in the respective legend. I.e. the last diagram presented is the most representative and could be used for this purpose.

Answer: We agree with the reviewer's comment, and we have modified the images (please see new Figures 2, 3, and 4) to make them clearer and more concise.

5. Lines 254-255: Which samples authors mean since they did not describe PMA/Ionomycin treatment in the materials and methods section, nor they presented any relative result in the respective figure?

Answer: We have inadvertently omitted the description of PMA/ionomycin and included this in the revised version. Please refer to the details in Answer 1.

6. Lines 262-263: Provide results by giving also SD or SEM.

Answer: We modified the presentation of results by using box plot figures, representing median and the three quartiles. This is now explained in legends of figures.

7. In the paragraph 2.2 of the results section the figures are not the right ones for the results described. Please correct it appropriately. Also, in order for authors to provide correlation among IFN- γ positive T cells and disease status and explain better their findings, the most appropriate is a correlogram and not a heat-map plot. Also, this paragraph needs to be written again. Specifically, authors should present firstly (%) of CD4 and CD8 T cells to show changes among stimulations, if there are any, secondly to present the MFI and then the iMFI. This could give more information regarding the different responses among the different clinical status of the groups used.

Answer: Multivariate and clustering analyses have been widely used to examine patterns of data variability, including in immunology studies. Although correlograms are useful for evaluating the correlation between numeric variables, they are less informative when categorical variables are required. Thus, it is necessary to subset the data before data analysis to incorporate categorical data into correlation analysis. In this case, it was necessary to calculate different correlograms for the different groups of variables. It is possible that we were not clear enough to explain this phenomenon. As shown in Figure 5, we tested whether we could detect patterns of distribution of the different cell population measurements according to the group they correspond to. In addition, the clustered heatmap has the additional advantage of showing a lower contribution of other variables, such as the nature of antigen stimuli, to the profile of responses. Therefore, we used an alternative version of the plot with traditional boxplots to reduce the values obtained from RPMI. The new version of the figures decreases the RMPI values for each data group. In addition, we changed the style of the plot from violin plots to boxplots.

8. Paragraph 2.3. Why did the authors detect T cell subpopulations in regard to different parasite serotypes? This is not clear. In addition, the T-cell stimulation process to discriminate different subpopulations is not described in detail in the respective paragraph of the Materials and Methods section. Also, have you tested IFN-gamma CD4 and CD8 T cells against serotype I, III and UND? Since this could give more sense to the results presented in this paragraph.

Answer: One main goal was to test whether the secondary immune response varied according to the infecting serotype within our groups. The stimulation conditions are described in the subsection on stimulus. We have included in this revised version a Table with the data on the percentage of CD4+ and CD8+ subsets induced after antigen stimulation according to the serotype (Table 1). In paragraph comprising lines 157 to 159, we write that the percentage of CD4+ and CD8+ T IFN γ -producing cells was unchanged according to infecting serotyping. We have included this explanation in the Discussion section.

9. A general comment is that T cells respond to NP10 stimulation in levels similar to antigen treatment. Have the authors excluded the spontaneous values (i.e medium only group) in order to detect the true value of response? Also, in lines 319-320, authors

support that "the response observed with NP10 can be attributed to the high concentration of NP inducing cell toxicity". If so, why did authors not use lower concentrations, since the responses detected in the presence of TE also could be attributed to NP and the antigen. This should be clarified.

Answer: We have addressed this in reviewer question #3. We used the RPMI group in all analyses presented, and the results of the culture medium alone were subtracted. As can be observed in the description of methods NP alone was used at 10 µg/ml but the stimulation of NP mixed with antigen was at 0.9 or 1.5 µg NP. This showed that NP at 10 µg/ml induced T cell producing IFN γ response. We included an explanation for this finding in the discussion section (lines 174 to 180). This data is important for future evaluation in humans to avoid these concentrations.

10. In line with the above comment, how do authors explain that NP alone induced CD8 memory cells (line 336)?

Answer: In relation with response to NP alone, we explain this best as antigen previous environmental exposure to maltodextrin that is part of probiotics. It has been demonstrated a CD8+ memory cells can be produced, please see at: Farhangi MA, Javid AZ, Sarmadi B, Karimi P, Dehghan P. A randomized controlled trial on the efficacy of resistant dextrin, as functional food, in women with type 2 diabetes: Targeting the hypothalamic-pituitary-adrenal axis and immune system. Clin Nutr. 2018 Aug;37(4):1216-1223. doi: 10.1016/j.clnu.2017.06.005. This reference was included (number 27).

11. The data show that there is not a significant difference in IFN-gamma producing T cells among naïve and infected individuals. Moreover, there has been non-specific production of IFN-gamma when PBMCs were stimulated with NPs. How do authors explain this? Is it due to the antigen or the carrier dose?

Answer: In the case of naïve individuals, memory T cells with different functions and phenotypes were observed in some uninfected people (seronegative) although theoretically this group did not have prior contact with the parasite. As they reside in endemic regions, previous contact with the parasite in low doses would explain these results, this phenomenon has been also reported in similar PBMC ex vivo experiments for Leishmania vaccine candidates in Brazil. In relation with NP, we explain this best as antigen previous environmental exposure to maltodextrin as described in previous answer.

General comment: The figures legends need to be more descriptive and the manuscript needs minor language editing.

Answer: We modified the legends of figures and checked language.

Reviewer #2 (Comments to the Authors (Required)):

This paper shows the results of an *ex vivo* study on the production of IFN - gamma by peripheral blood mononuclear cells from not immune or chronically infected or with chorioretinitis due to *Toxoplasma gondii* individuals after stimulation with a *T.gondii* total antigen-loaded in maltodextrin nanoparticles NPL/TE at different concentration

as a preliminary study for a vaccine. Furthermore, the immunophenotyping of CD4 and CD8 cells and the serotyping of infecting parasite were performed. The authors found that the immunological response after stimulation with NPL/TE is different in different individuals and this is not a consequence of different genotypes of *T. gondii* infection and that problem with ocular toxoplasmosis have a different and stronger response to the antigen stimulation. They conclude that NPL/TE conjugate could be an excellent vaccine candidate. In my opinion the study is well conducted. The results are clearly expressed but must be stressed that they are really preliminary data due to the low number of tested samples. I also want to ask to specify why the incubation time was chosen at 72 h. The results of serotyping must be described in the text also. Line 70-71 there is a repetition

Answer: We thank the reviewer for the positive comments and have included this limitation due to the low number of samples in the discussion section. We included the reason why we chose 72 h: The time of 72 h was chosen after preliminary experiments showed that in other conditions, no significant production of IFN- γ was obtained. We have also included a discussion of the serotypes. One of the main goals was to test whether the secondary immune response varied according to the infecting serotype within our groups. We have included this in the Results section. In paragraphs comprising lines 157 to 159 it is explained how the percentage of CD4+ IFN γ producing cells was unchanged according to infecting serotyping. Repetition has been deleted in line 70.

Reviewer #3 (Comments to the Authors (Required)):

1. One issue is high IFN-g productions following stimulation with NP.

Answer: NP alone was used at 10 $\mu\text{g/ml}$ but the stimulation of NP mixed with antigen was at 0.9 or 1.5 μg NP (lanes 315 to 317). This showed that NP at 10 $\mu\text{g/ml}$ induced T cell producing IFN γ response. We included an explanation for this finding in the discussion section (lanes 174 to 180). This data is important for future evaluation in humans to avoid these concentrations.

Another issue is large variations among individuals within each of the three groups of the individuals.

Answer: This is work with human samples, where a large variation is expected. The interest of the present work is precisely to afford data describing the immune response in the human population facing a conjugate that aims to become a vaccine candidate. We included explanation about this in lines 136 to 139.

In addition, after stimulation with NP/TE, there were no statistically significant differences in IFN-production between asymptomatic and ocular toxoplasmosis individuals, indicating that this assay would not be helpful for the diagnosis of ocular toxoplasmosis.

Answer: This is in line with our previous findings on this group of people. Previous work of our group showed that it is important to analyze the IFN γ /IL10 ratio but not the absolute values of cytokines production.

The authors also analyzed T cell subpopulations that produce IFN-g in response to these four antigen preparations. However, there are no clear differences in the frequencies of IFN-g+ cells in each of 6 subpopulations of the T cells among the three groups of the individuals and among four different antigen preparations (Figure 5). Although the authors state in the Abstract that "We found the populations with higher IFN-g were CD8+ central memory T cell subset in chronic asymptomatic infection and of CD4+ central memory cell and terminally differentiated effector memory subset from people with ocular toxoplasmosis", the results shown in Figure 5 do not support this statement. Overall the quality and significance of the data show in in this manuscript are not high.

Answer: We reanalyzed all data and we reformulate conclusions. The value of the present work is to precisely show how can be the individual variation in these preparations. This is a pioneer work analyzing complex *ex vivo* human subset of T cell response face a vaccine candidate for toxoplasmosis. We modified the abstract and conclusions.

We would like to thank the reviewers for their thoughtful comments, which have helped improve the manuscript. We hope now that our paper will be ready to be accepted in Journal of Infection and Public Health

Yours sincerely,

Jorge Enrique Gomez Marin, MD, PhD

Director Grupo GEPAMOL

Centro de Investigaciones Biomedicas Universidad del Quindio

April 12, 2024

RE: Life Science Alliance Manuscript #LSA-2023-02486-TR

Dr. Jorge Enrique Gomez Marin
University of Quindío
Centro de Investigaciones Biomedicas
Carrera 15 Calle 12 Norte
Armenia 630004

Dear Dr. Gomez Marin,

Thank you for submitting your revised manuscript entitled "Human T cell activation with Toxoplasma gondii antigens loaded in maltodextrin nanoparticles.". We would be happy to publish your paper in Life Science Alliance pending final revisions necessary to meet our formatting guidelines.

- please address Reviewer 1's remaining comments
- please be sure that the authorship listing and order is correct

A. FINAL FILES:

B. MANUSCRIPT ORGANIZATION AND FORMATTING:

****It is Life Science Alliance policy that if requested, original data images must be made available to the editors. Failure to provide original images upon request will result in unavoidable delays in publication. Please ensure that you have access to all original**

data images prior to final submission.**

The license to publish form must be signed before your manuscript can be sent to production. A link to the electronic license to publish form will be available to the corresponding author only. Please take a moment to check your funder requirements.

Sincerely,

Reviewer #1 (Comments to the Authors (Required)):

In the revised form of the manuscript authors have answered satisfactorily in the majority of concerns. There are some minor considerations that should be solved:

1. Authors should present the numerical values of IFN γ production detected via ELISA for all stimuli in Table, so it will be clearer the difference among groups and stimulations.
2. How authors explain the difference seen in results among IFN γ iMFI CD4+/CD8+ and the IFN γ in the CD4+/CD8+ T cell subsets, since in the first case there was not any difference among groups, despite the fact that authors present them as statistical different.
3. Also, authors should add a Table with the characteristics of the volunteers entered the study.

Reviewer #1 (Comments to the Authors (Required)):

In the revised form of the manuscript authors have answered satisfactorily in the majority of concerns. There are some minor considerations that should be solved:

1. Authors should present the numerical values of IFN γ production detected via ELISA for all stimuli in Table, so it will be clearer the difference among groups and stimulations.

Answer: We included one new Table to show the numerical values of IFN γ levels.

2. How authors explain the difference seen in results among IFN γ iMFI CD4+/CD8+ and the IFN γ in the CD4+/CD8+ T cell subsets, since in the first case there was not any difference among groups, despite the fact that authors present them as statistical different.

Answer: We included in discussion section the next explanation (lines 170 to 172): "This difference was not evident when the IMF of CD4+ and CD8+ T cells was analyzed probably due to the intrinsic differences in the effector response to antigens by distinct CD8+ and CD4+ T effector cell types".

3. Also, authors should add a Table with the characteristics of the volunteers entered the study.

Answer: We included a new Table with characteristic of volunteers. We would like to thank the reviewer for their thoughtful comments, which have helped improve the manuscript. We hope now that our paper will be ready to be accepted.

April 16, 2024

RE: Life Science Alliance Manuscript #LSA-2023-02486-TRR

Dr. Jorge Enrique Gomez Marin
University of Quindío
Centro de Investigaciones Biomedicas
Carrera 15 Calle 12 Norte
Armenia 630004
Colombia

Dear Dr. Gomez Marin,

Thank you for submitting your Research Article entitled "Human T cell activation with Toxoplasma gondii antigens loaded in maltodextrin nanoparticles.". It is a pleasure to let you know that your manuscript is now accepted for publication in Life Science Alliance. Congratulations on this interesting work.

DISTRIBUTION OF MATERIALS:

Again, congratulations on a very nice paper. I hope you found the review process to be constructive and are pleased with how the manuscript was handled editorially. We look forward to future exciting submissions from your lab.

Sincerely,
